# Anxiety symptoms and coping strategies used by older adults during COVID-19: A national e-study of linkages among and between them

Gail Low[1]☯*, Alex B. França[2]☯, Zhiwei Gao[3]‡, Gloria Gutman[4]‡, Sofia von Humboldt[5]‡, Hunaina Allana[1]‡, Donna M. Wilson[6]‡, Anila Naz[6]‡

1 Faculty of Nursing, MacEwan University, Edmonton, Canada, 2 Laboratory of Human Development and Cognition, Federal University of São Carlos, São Paulo, Brazil, 3 Division of Community Health and Humanities, Memorial University of Newfoundland, St. John's, Canada, 4 Gerontology Research Centre, Simon Fraser University, Vancouver, Canada, 5 William James Center for Research, ISPA—Instituto Universitário, Lisbon, Portugal, 6 Faculty of Nursing, University of Alberta, Edmonton, Canada

☯ These authors contributed equally to this work.
‡ ZG, GG, SH, HA, DW, and AN also contributed equally to this work.
* lowg4@macewan.ca

## Abstract

A global pandemic is a hardly typical and anxiety-dampening event. Research in the first year of the COVID-19 pandemic tells of associations between advancing age and anxiety dampening. The aim of this study was to further investigate this by examining and creating a blueprint of older Canadians' symptoms of pandemic-related anxiety and coping strategies, and linkages among and between them. A national e-survey was conducted in the second year of the pandemic with 1,327 older Canadians, when national public health measures lifted. Anxiety symptoms were measured using the Geriatric Anxiety Scale - 10. Participants also completed the Coping with Stress and Anxiety personal assessment tool. Network Analyses revealed a troubling trio of anxiety symptoms of central importance to our respondents: feelings of restlessness, muscle tension and having no control over their lives. Restless and no control over my life explained between 64–68% of the variance in 8 other anxiety symptoms. Coping seemed to occur through trial and error. Some strategies appeared to work in tandem and others in opposition to each other. Remembered resilience and staying active functioned as bridges shielding older people from worry, restlessness, and tension through spurning other remedial actions. This study provides evidence of a stable and predictable network of anxiety symptoms containing three particularly pernicious symptoms and the complex and arduous nature of mentally healthy recovery work. A visual representation of how anxiety symptoms can operate as a network might help older people better understand their own symptom experiences. Combining the two networks offers a blueprint of what within-person recovery might look like and a visual teaching tool for practitioners and program developers; older people could gain added insight into their own recovery experience.

which permits unrestricted use, distribution, and reproduction in any medium, provided the original author and source are credited.

**Data availability statement:** There are ethical restrictions on the data being analyzed in this study. The data are not publicly sharable data. In our Study Information and Implied Consent Letter approved by the University of Alberta Human Research Ethics Board (Pro0118512) and shared with potential e-survey responders, we stated the following: "Confidentiality and Anonymity: The information that you will share will remain strictly confidential and will be used solely for the purposes of this research. The only people who will have access to the research data are the four research team professors from the University of Alberta, Simon Fraser University and Memorial University) and the research assistant helping them to analyze surveys and to generate findings and summaries of findings for recipe book development. Your answers to open-ended questions may be used verbatim in presentations and publications but neither you (nor your organization) will be identified." Mental health is a highly sensitive matter and remains a social taboo in Canadian society (https://www.camh.ca/en/driving-change/addressing-stigma; also see https://cmhadurham.ca/stigma-and-discrimination/). Our e-survey responders were sharing information about their mental health and at a very frightening time. They were transitioning into larger open spaces, with COVID-19 still lingering and being most at risk for COVID-19-related hospitalizations and dying. Our e-survey responders consenting to participate and to have our findings published in aggregate form did so with the understanding that our Confidentiality and Anonymity pledge would be kept. We must ensure that data are shared in accordance with participant consent. The Canadian Nurses Association Code of Ethics (https://www.cna-aiic.ca/en/nursing/regulated-nursing-in-canada/nursing-ethics) requires that my everyday research practices are practices "Promoting and Respecting Informed Decision-Making." For these good reasons, fellow researchers are asked to send data inquiries and requests to: REB@macewan.ca.

**Funding:** This study was supported by the RTOERO Foundation Research Grant in the form of salaries [RES0056223 to HA and AN].

**Competing interests:** The authors have declared that no competing interests exist.

## Introduction

A pandemic is an unprecedented event, typically not falling within the everyday scope of experiences of the public [1,2]. In the first year of the COVID-19 pandemic, governments worldwide were asked to activate emergency response mechanisms, including keeping a distance from others for one's own physical health preservation [3,4]. While one of the most frightening consequences of large-scale traumatic events are threats to people's physical safety [5], mental health threats also occur; some of both types of threats are disproportionate for certain subgroups. For example, during COVID-19, while older people were less likely than younger or midlife adults to experience family [6] and financial [7] discord, they were most at risk for infections, hospitalizations, and death [8].

With mental health harms being a realistic secondary consequence of COVID-19 [9], Canadians were deluged with an expert-driven "flood of wellness and self-care information", most of it virtually delivered [10]. It appears however that while a minority of older Canadians were accessing e-support [11], others exercised their inner capacity for resourcefulness rather than seeking an expert rescue fix [12].

Subsequent research tells us older people rated their mental health and well-being significantly higher than midlife adults [6,7,13,14] and younger adults [15–17]. Still others were happier and more satisfied with life [18,19], and less anxious [20–22] and depressed [19,22,23] than midlife adults. However, older people also spoke of emotional meltdowns over health service delays [24], feeling lonely and without a companion [6], and frustrated over having idle 'capable of helping others' hands [13,25].

Qualitative researchers drew attention to older people's accounts of a variety of coping strategies, while survey researchers estimated their impacts, usually assessing one or a few strategies at a time. In Hong Kong, older people were reported to have prioritised time and energy for family and friends [18]. Portuguese age contemporaries persevered enough to appreciate e-connectedness [24]. Older Brits welcomed less busyness in life and took comfort in routine activities [26]. Older Canadians exercised outside [25]. Older Americans found solace in connecting with close friends [27]. Older Italians gained some satisfaction from doing housework [28]. For older German people, this meant opting for quality social interactions, not spreading themselves too thin [19]. In Canada, some older people found seeking credible COVID-19 information [29], leisure activities [30], and eating and sleeping well [31] mentally beneficial.

What is needed now is research that will help us to better understand what optimal structures for mentally healthy living in pandemic times might look like and what lessons have been learned that can be applied when faced with future pandemics [32]. In the first year of COVID-19, some studies pointed to the merits of sticking to past coping strategies to protect one's mental health [18,33]. In a pandemic, people's health and social circumstances can also quickly shift and demand unanticipated concomitant remedial actions [34]. COVID-19 has trickled into many nooks and crannies of people's lives and, for example, through the loss of loved ones and disrupted everyday routines, and posed threats to their physical health, and social and economic livelihoods [1,35] and in simultaneity [36].

We know little about optimal structures for mentally healthy living, particularly at key points in the COVID-19 pandemic. In the first year, we learned of older people pivoting away from a familiar community gym to the novelty of an at-home art easel [25] and mapping out what they thought a post-pandemic social life might look like [13]. With the lifting of public health measures, many community organisations have evolved from being completely virtual into a mix of face-to-face and virtual support. Older Canadians rating these options have indicated that they generally prefer in-person or phone contact [8].

During an unprecedented pandemic, people are likely to use a variety of strategies and, perhaps through a process of trial and error, stay mentally healthy and well [37]. The aim of this study was to create a visual representation of older Canadians' symptoms of anxiety experiences and another of the linkages between their most pernicious symptoms and the strategies they used to cope with them. Knowing how contemporaries experienced and managed their anxiety could motivate others into action and give them a sense of hope [34,38]. Good COVID-19 research supports mentally healthy recoveries within and between people [32].

## Materials and methods

### Ethics approval and consent to participate

This project is approved by the University of Alberta [Pro00118512_REN3] and MacEwan University [File No. 102407] Human Research Ethics Board. All participants gave informed consent to participate and to publish these findings prior to filling the survey, online. All methods were carried out in keeping with the Declaration of Helsinki.

### Participants and procedures

Survey data were collected using an e-survey co-designed with Qualtrics and launched between July 1st, 2022, and August 15, 2022. E-survey responders were recruited through a study advertisement sent to a survey panel of older people in all 10 Canadian provinces. We used stratified sampling to recruit older people similar in proportion to the Canadian population in terms of their age group and sex-at-birth, and their level of education [39]. Strata categories were Canadian Census categories [40].

Before taking our completely voluntary e-survey, advertisement responders were taken to the Qualtrics survey platform to view a detailed study information letter. This letter fully informed advertisement responders about our study rationale and aim, what they were being asked to do, and with assurances of confidentiality and 24/7 mental health support contact information. Advertisement responders who viewed this letter and answered 'YES' to taking part in the e-survey were assigned a personal e-survey link and password. All such informed consent providers were awarded a total of $4 in consumer points that they could put towards everyday essentials like coffee or gas, regardless of the number of e-survey questions that they had chosen to respond to.

### Measures

Our computer and mobile phone friendly e-survey took on average, 11.76 minutes to finish, give or take 1.10 minutes. Along with age, sex, and education, responders were asked their home province, and health and marital status [40].

Responders rated their anxiety using the Geriatric Anxiety Scale or GAS-10 [41,42]. This 10-item survey tool exhibited strong internal consistency (Cronbach's α = .921). Item examples are 'I was irritable' and 'I could not control my worry'. Each item is rated on a 4-point Likert scale (0 = 'not at all' to 3 = 'all of the time').

Responders also filled out a Coping with Stress and Anxiety [43] personal assessment tool (Cronbach's α = .747). This tool for public consumption helps people take stock of the everyday strategies that they were using to help them transition into open spaces. Responders answered 'YES' or 'NO' to whether they used each strategy. This particular tool was after the GAS-10. Responders moved from reflecting on their anxiety symptoms to thinking about their capacity for managing or mitigating them.

PLOS Mental Health

## Statistical analysis

To investigate the relationship between the strategies captured by the Coping with Stress and Anxiety personal assessment tool and anxiety symptoms as measured by the GAS-10, we constructed two networks. The first network exclusively incorporated symptoms of anxiety from the GAS-10, with a view to identifying the most central or pernicious symptoms. A subsequent network analysis combined responders' most pernicious GAS-10 symptoms and with their Coping with Stress and Anxiety strategies.

## Network estimation

Our networks were explored using a Gaussian Graphical Model. We estimated the network by applying the Least Absolute Shrinkage and Selection Operator (LASSO) regularisation technique to control for spurious interactions. The LASSO parameter was selected through the Extended Bayesian Information Criterion (EBIC), with the hyperparameter gamma set to the default value of.50 for model selection [44]. Nodes represent questionnaire items. In our case, we would have 10 symptoms of anxiety nodes and 16 coping behaviour nodes. Relationships between anxiety symptoms and between anxiety and coping symptoms are represented by edges. The weight of the edges represented the strength of association between two variables after controlling for all others in the network.

## Network description and node characteristics

We evaluated the relative importance of nodes representing GAS-10 anxiety symptoms using Strength Centrality, which is the sum of all the weights connected to a symptom in absolute value. This index tells how well connected a particular anxiety symptom is to all other nodes or GAS-10 symptoms in the network, meaning they play a key role in maintaining and potentially exacerbating anxiety symptomatology [45]. In line with previous studies, we focused our interpretation on symptoms with the highest centrality values rather than on individual edge strengths, as this approach is commonly used in empirical network analysis research [46]. Strength Closeness and Betweenness were not reported because neither have generally shown stability and so they can be problematic measures since they treat associations as distances [44,47].

In our combined symptoms-coping network, Bridge Strength (i.e., the sum of the absolute value of all edges between a GAS-10 symptom and a Coping with Stress and Anxiety behavior) was estimated. Nodes that best foster interconnections between symptoms and coping networks possess higher Bridge Centrality scores. If 'seeking support from loved ones' had high bridge centrality, much like a bridge connecting two lands separated by a body of water, seeking support would connect responders' most pernicious GAS-10 symptoms with other possibly beneficial coping behaviours.

We also generated a Predictability Index. Predictability is the variance in a node as explained by all other nodes in the network. In a network plot, ring-shaped pie charts represent predictability [48]. High predictability in a GAS-10 network would indicate that responders' anxiety is primarily explained by the presence of strong interactions between the symptoms within that network, whereas low predictability is indicative of there being other factors missing from this GAS-10 network [48].

The robustness of Strength Centrality and Predictability was examined by calculating indices of stability. Stability within our GAS-10 network and our combined network were estimated using a case-dropping bootstrap procedure. We focused on the Correlation Stability Coefficient (CS-coefficient), the proportion of the sample that could be dropped while still retaining with 95% probability a correlation between the index computed on the full sample and the index computed on the reduced sample. A value below 0.25 indicates insufficient stability; a value larger than 0.50 is preferred [44].

All statistical analyses were performed using RStudio 2023.06.1 [49], and packages bootnet version 1.5.3 [44], and graph version 1.9.3 [50].

## Results

A total of 1,327 "YES" responders took our e-survey. They were strikingly similar in terms of age, sex, and education to the general population of older Canadians' (Table 1). The coping strategies they identified with are shown in Table 2.

**Table 1. Sociodemographic characteristics (N = 1,327).**

| Variables | n | %[a] |
|---|---|---|
| **Age** | | |
| 60–64 | 384 | 28.9 |
| 65–69 | 335 | 25.2 |
| 70–74 | 264 | 19.9 |
| 75–79 | 131 | [b]9.9 |
| 80–84 | 145 | 10.9 |
| 85+ | 61 | [b]4.6 |
| **Sex at birth** | | |
| Female | 671 | [b]50.6 |
| Male | 620 | 46.7 |
| **Marital status** | | |
| Widowed/not living common law | 191 | 14.4 |
| Divorced/separated/not living common law | 217 | 16.4 |
| Living common law | 89 | 6.7 |
| Married | 671 | 50.6 |
| Never married/not living common law | 154 | 11.6 |
| **Gender** | | |
| CIS gender woman | 502 | 37.8 |
| CIS gender man | 516 | 38.9 |
| Transgender/non-binary | 178 | 13.4 |
| **Province** | | |
| British Columbia | 226 | 17.0 |
| Prairie Provinces | 312 | 23.5 |
| Ontario | 510 | 28.4 |
| Quebec | 148 | 11.2 |
| Maritime Provinces | 129 | 9.7 |
| **Education status** | | |
| University certificate, diploma, degree below/at bachelor level | 275 | 20.7 |
| College, CEGEP or other non-university certificate or diploma | 238 | 17.9 |
| Apprenticeship, trades certificate or diploma | 168 | 12.7 |
| Secondary (high) school or equivalent | 392 | 29.5 |
| No degree, certificate, or diploma | 251 | [b]18.9 |
| **Perceived health** | | |
| Poor/fair | 407 | 30.7 |
| Good | 604 | 45.5 |
| Very good/Excellent | 298 | 22.5 |
| **Geriatric Anxiety Scale** | | |
| I am not at all anxious | 273 | 20.6 |
| I feel minimal anxiety | 513 | 38.7 |
| I feel mildly anxious | 136 | 10.2 |
| I feel moderately anxious | 70 | 5.3 |
| I feel severely anxious | 280 | 21.1 |

[a]Frequency proportions do not sum to 100% due to missing values.

[b]Frequency proportions falling below Statistics Canada Census Proportion.

**Table 2. Self-identified coping strategies from stress and anxiety checklist[a] (N = 1,327).**

| | Yes | | No | |
|---|---|---|---|---|
| | n | %[a] | n | %[a] |
| I accepted that some fear and anxiety is normal | 1163 | 87.6 | 142 | 10.7 |
| I sought credible information about COVID-19 | 581 | 43.8 | 718 | 54.1 |
| I found a balance staying tuned in and knowing when to take a breather | 1016 | 76.6 | 289 | 21.8 |
| I brought an internal mindset to unplugging | 281 | 21.2 | 1021 | 76.9 |
| I dealt with problems in a structured way | 1020 | 76.9 | 281 | 21.2 |
| I remembered that I am resilient/careful with what If's | 989 | 74.5 | 315 | 23.7 |
| I challenged anxious worries and thoughts | 841 | 63.4 | 460 | 34.7 |
| I decreased other sources of stress in my life | 756 | 57.0 | 544 | 41.0 |
| I practiced relaxation and meditation | 503 | 37.9 | 796 | 60.0 |
| I sought support from loved ones | 565 | 42.6 | 735 | 55.4 |
| I was kind to myself | 1107 | 83.4 | 198 | 14.9 |
| I ate healthy | 950 | 71.6 | 352 | 26.5 |
| I avoided substance use | 854 | 64.4 | 448 | 33.8 |
| I had moderate caffeine intake | 976 | 73.5 | 328 | 24.7 |
| I got proper rest and sleep | 1012 | 76.3 | 290 | 21.9 |
| I stayed active | 914 | 68.9 | 389 | 29.3 |

[a]Frequency proportions do not sum to 100% due to missing values.

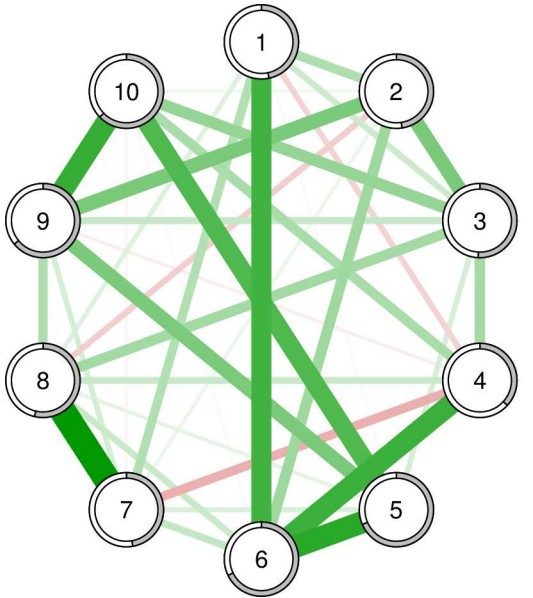

1: Irritable
2: Isolated
3: Dazed
4: Hard time sitting still
5: Not control my worry
6: Restless
7: Tired
8: Muscle tension
9: No control over my life
10: Apprehensive

**Fig 1. Anxiety symptoms network (n = 1,224).**

## Anxiety symptoms network

The CS Coefficients for Strength (CS = 0.60) and Predictability (CS = 0.95) indicated that the GAS-10 anxiety symptoms network was stable. This stable GAS-10 network had 37 non-zero edges out of 45 possible edges (Fig 1). Especially strong connections emerged between Node 1 ('*Irritable*') and Node 6 ('*Restless*'), and between Node 5 ('*Not control my*

*worry*') and Node 4 ("*Hard time sitting still*"). Strong connections were also observed between Node 7 ('*Tired*') and Node 8 ('*Muscle tension*'), and between Node 9 ('*Had no control over my life*') and Node 10 ('*Apprehensive*'). Other connections, such as between Nodes 1 and 10, were absent. This missing connection suggests that apprehension and irritability are statistically independent sources of bother, in their own right [51].

Ring-shaped pie charts represent predictability. The nodes are arranged in order according to the legend and represent anxiety symptoms (GAS-10 items). The size and density of edges between nodes represent the strength of connectivity.

As shown below in Fig 2, the three symptoms with the highest standardised Strength Centrality in the GAS-10 anxiety symptoms network) were '*Restless*', '*No control over my life'* and '*Muscle tension*'. In terms of mean Predictability, 55% of the variance in all GAS-10 nodes was explained by neighbouring symptoms. Symptoms with the highest Predictability were Node 5 ('*Not control my worry*'; 68%), Node 6 ('*Restless*'; 68%), and Node 9 ('*No control over my life*'; 64%).

### Combined anxiety symptoms and coping strategies network

A visual representation of the combined network is shown in Fig 3. As indicated by CS coefficients for Strength (CS = 0.85) and Predictability (CS = 0.85), the combined anxiety symptoms and coping strategies network is stable. Overall, the combined network shows clear associations both within and between Coping with Stress and Anxiety strategies and GAS-10 symptoms of Anxiety. The mean Predictability across all nodes in this combined network was 21%. Node 1 ('*Restless'*), Node 2 ('*Muscle tension'*), and Node 3 ('*No control over my life'*) were strongly positively connected. Across this network, Node 3 ('*No control over my life'*) had the highest predictability (49%).

Ring-shaped pie charts represent predictability. The nodes are arranged in order according to the legend and represent anxiety symptoms and coping strategies. The size and density of edges between nodes represent the strength of connectivity. Blue edges are positive relationships, and red edges are negative relationships.

Strong positive connections emerged between Node 12 ('*Relaxing and meditating'*) and Node 13 ('*Seeking loved ones' support'*), and between Node 5 ('*Seeking credible information'*) and Node 6 ('*Tuning in and breaking away from news'*). Other strong connections were between Node 15 ('*Eating healthy'*) and Node 19 ('*Staying active'*). There were negative connections between, for example, Node 14 *('Kind to myself'*) and Node 5 ('*Seeking credible information'*), and Node 15 ('*Eating healthy'*) and Node 6 ('*Tuning in and breaking away from news'*).

We also analysed Bridge Strength (Fig 4) among the combined networks of symptoms and coping strategies. In this respect, Node 9 ('*Remembering self as resilient'*) and Node 19 ('*Staying activ*e') were particularly pivotal strategies. The importance of both such strategies is further depicted in Fig 3 as a positive association between Node 9 ('*Remembering self as resilient'*) and Node 2 ('*Muscle tension'*). Negative associations emerged between Node 9 ('*Remembering self as resilient'*) and Node 3 ('*No control over my life'*), and between Node 19 ('*Staying activ*e') and Node 3 ('*No control over my life'*).

## Discussion

This study employed Network Analysis to create a visual representation of older Canadians' symptoms of anxiety experiences and another of the linkages between their most pernicious symptoms and the strategies they used to cope with them. Novel discoveries of linkages among and between them are summarised and discussed below.

### Network of anxiety symptoms

In the first year of COVID-19, older Canadians were generally least prone to high anxiety [8,20,21,52]. The loss of local contemporaries can make the threat of physical health harms more self-relevant [5], let alone on a global level [53]. This may partially explain the observed strong connections between anxiety symptoms (e.g., '*Irritable*' and '*Restless*', '*Not control my worry*' and '*Hard time sitting still*', '*Tired*' and '*Muscle tension*', '*No control over my life*' and '*Apprehensive*'). These pairs might represent clusters wherein one symptom can exacerbate the other or they might arise from common

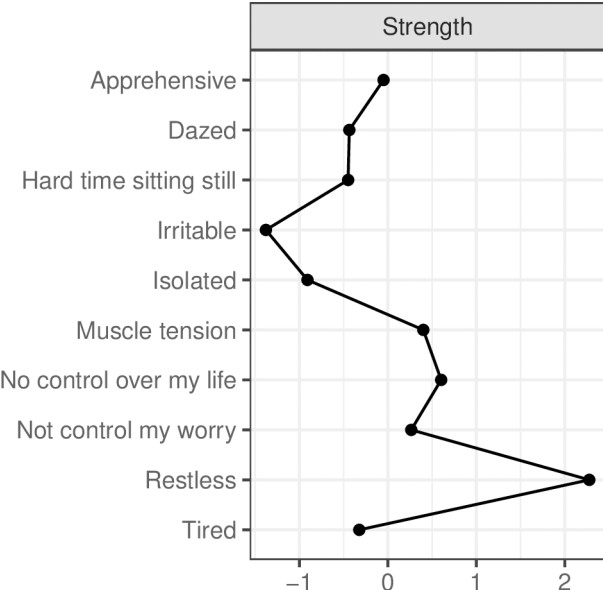

**Fig 2. Strength centrality of GAS-10 symptoms, shown as standardised z-scores.**

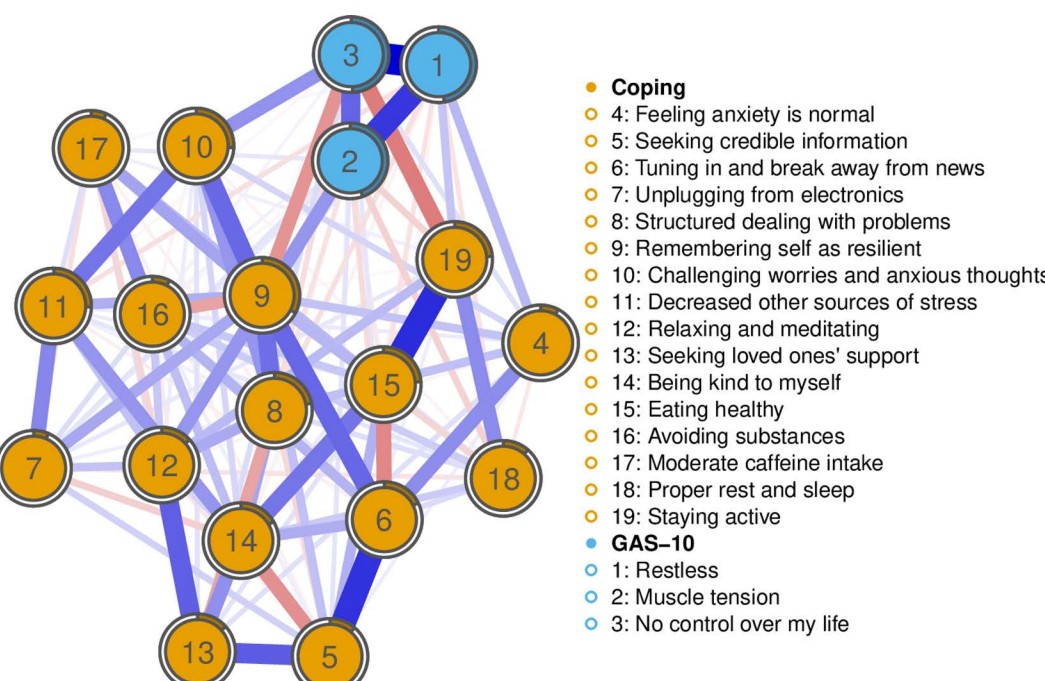

- **Coping**
  - 4: Feeling anxiety is normal
  - 5: Seeking credible information
  - 6: Tuning in and break away from news
  - 7: Unplugging from electronics
  - 8: Structured dealing with problems
  - 9: Remembering self as resilient
  - 10: Challenging worries and anxious thoughts
  - 11: Decreased other sources of stress
  - 12: Relaxing and meditating
  - 13: Seeking loved ones' support
  - 14: Being kind to myself
  - 15: Eating healthy
  - 16: Avoiding substances
  - 17: Moderate caffeine intake
  - 18: Proper rest and sleep
  - 19: Staying active
- **GAS–10**
  - 1: Restless
  - 2: Muscle tension
  - 3: No control over my life

**Fig 3. Top 3 GAS-10 symptoms and coping with Stress and Anxiety strategies network.**

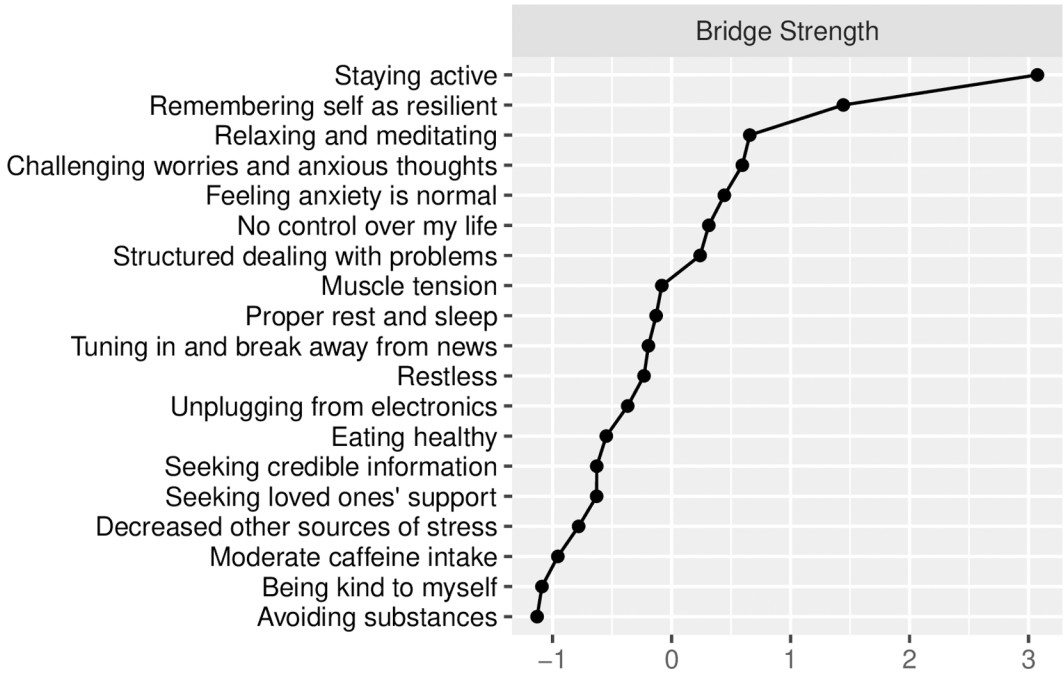

**Fig 4. Bridge strength centrality shown as standardised z-scores.**

underlying factors. No one is immune to unfavourable shifts in anxiety [34]. This past year alone, the lifetime prevalence of pernicious anxiety among older adults has doubled (5.2% to 13.3%) and 1 in 3 report unmet mental health needs [8].

Our respondents were moving into opened spaces while still living most at risk for becoming infected with and dying from COVID-19. This is a rather precarious position to be in. Central to their precarity were feelings of restlessness, muscle tension, and of having no control over their lives. Restlessness and no control were particularly pernicious, as they explained 64% to 68% of the variability in 8 other anxiety symptoms. Learning of their connectedness from age-peers in similar circumstances could be a source of good mentally healthy recovery work [9,54]. Much like long-COVID [40], pernicious anxiety can impair our everyday functional ability and be a source of emotional distress [8]. Strategies to disempower this trio are thus pivotal.

## Combined network of anxiety and coping strategies

We examined connections between older Canadians' restlessness, muscle tension, and perceived lack of control over their lives, and mental and behavioural strategies to manage or mitigate this troubling trio. This combining helped us build a blueprint showing what within-person recovery might look like for other older people.

## Positively connected coping strategies

Three sets of strategies were positively connected and appeared to be used in tandem, with one spurning on a significant other.

The positive association between '*seeking credible information*' and '*tuning in and breaking away from news*' [5] remind us that unbalanced information about hypothetical sources of harm or displayed images of threat in the news can undermine peoples' sense of safety. COVID-19 evoked uncontrolled and near constant streams of news [54]. For older people in particular, public discourse through social media, popular press, and government official messaging set the stage for

being characterised as vulnerable COVID-19 carriers [55,56]. Those indirectly affected through media exposure could personalise it and/or see illness potential in themselves [57].

Our associations suggest that credible sources might curb people seeking out news elsewhere. This proactive constructive approach is important given that assumptive or misinformed sources have a penchant for fuelling, not quiescing, anxiety [58,59]. Credible sources such as organisational officials can stunt rumours and encourage return information seeking [60]. In the first year of the pandemic, through Canada.ca, all Canadians were strongly encouraged to limit everyday news media exposure, even from authoritative sources. The linkage between seeking credible information and COVID-19 news suggests that our respondents heeded such advice.

This observed pairing of '*relaxing and meditating*' and '*seeking loved ones' support*' could have been a means for our respondents to avoid or mitigate family conflicts or perhaps mask their own COVID-related worries. Meditation has been found to be used more by older Canadians with no or minimal to mild symptoms of anxiety [61]. While some research [33] indicates some older people made strategic use of their resources to lessen loneliness and isolation and some [25] describe 'pivoting' or substituting pre-COVID routines with novel activities to stay mentally well, our responders did so through strategic pairing, with in tandem '*eating healthy' and 'staying active*' perhaps being another means by which respondents further minimise loved ones' stress. In view of this, older Canadians are of generous spirit.

## Negatively connected coping strategies

Negative associations suggest that coping behaviours to manage or mitigate anxiety can also work in opposition. One such case-in-point is *'being kind to yourself' and 'seeking credible information'*.

In the first year of COVID-19, misleading, false, or inaccurate information about COVID-19 was found to increase older Canadians' anxiety [61]. A recent literature review also linked being privy to risk-related COVID-19 news with poorer mental health [62], and with higher symptoms of anxiety [63,64]. Others' unhappiness and symptoms of depression seems to be associated with news following frequency [65]. Sharing negative news appears to trigger anxiety [66]. In the context of this study, being kind to yourself through shielding yourself from distressing facts represents another constructive approach to moving back into open social spaces after two years of pandemic restrictions.

There was a negative association between '*tuning in and breaking away from news' and 'eating healthy'*. We could not find any published studies about older Canadians' eating habits in relation to COVID-19 news following. In the first year of COVID-19, some older Canadians named [67] and found [20,31] eating healthy food mentally beneficial. A meta-analysis across some 40 countries, including Canada [68], revealed associations between unhealthy eating behaviours and anxiety in individuals between 18 and 74 years of age, albeit more so in younger age groups. In one study [6], older Canadians found not having or not knowing how to access food particularly distressing. As with 'being kind to myself', these are cross-sectional associations. Future qualitative research is warranted to better understand these relationships. We wondered whether older people were more attuned to the consumption of pandemic versus pantry staples. Prevailing news narratives centred around their vulnerability [56]. On the other hand, eating healthy might be a distraction, dulling one's appetite for news policing or for any news at all.

## Implications of findings

The positive and negative associations observed in our combined network may be explained by people managing or mitigating anxiety through a process of trials and errors [37]. Resilient people are flexible people with an ability to reframe how they think, act, and feel about what is in front of them, whether this concerns precarious health, work, and/or family matters [69]. In the first year of COVID-19, resilient older people appeared to be better able to weather its many impacts [35,70]. Qualitative [71] and quantitative [23,31] researchers partly attributed more favourable mental health and well-being of older people to personal resilience.

'*Remembering self as resilient*' was an asset for our respondents on several fronts when they were moving into open spaces. Such recollections were negatively associated with feeling a '*lack of control over my life*'. Much like in the first pandemic year, positive reframing circumvented the negative impacts that intolerable uncertainty had on anxiety [72]. Being wise to life's physical uncertainties and open-minded has also been helpful to older people well before COVID-19 [73,74].

Resilience was also important because a '*lack of control over my life*' seemed to either deter '*staying active*', or vice versa. Resilient older people are likely to be flexible about how often they stay active and what they gain from it. Simply getting outside has also brought older people closer to nature [75] and has given them some kind of social life [33] and helped them feel less isolated in general [71]. This kind of reframing has led others to find contentment with small social circles [19,61] and in befriending technology [24]. Again, both scenarios are plausible. Observed relationships in a cross-sectional Network Analyses cannot tell us which variable is the true instigator [50,76].

The same can be said of respondents' remembering resilience strongly shielding them from feeling restless, tense, and as if they had no control over their lives. This troubling trio may trigger memories of overcoming other hardships. Future network analyses should include a measure of resilience such as the Brief Resilience Scale [77] to shed further light on its associations with symptoms of anxiety and strategies to manage or mitigate them. Programs or interventions to bolster resilience might benefit older people feeling restless, tense, and as if they have no control over their lives. Ideally, interventions strengthen resilience and activity propensities.

Network analysis offers advantages that traditional multivariate analytical methods like logistic regression do not. Network analysis examines variable associations simultaneous with, rather than in isolation of one another [76]. People are not likely to experience one symptom of anxiety at a time, but rather a host of symptoms [78]. Anxiety symptoms most closely affiliated with one another can also be identified beside each other for subsequent testing in networks with other variables [76]. Older Canadians in this study engaged in multiple remedial strategies to manage a troubling trio of anxiety symptoms. The presence and intensity of these symptoms may also influence the selection and perceived effectiveness of these coping strategies, suggesting reciprocal rather than strictly unidirectional relationships. Network analysis embraces and offers nuanced understandings of the complexities of mental health recovery work [50,76]. Good COVID-19 research helps everyday people, practitioners, and program planners take stock of what others have learned at key points in time [32].

Taken together, the findings of this study tell us that mental health recovery work requires a wide variety of coping strategies, with some working in tandem and some in opposition. Symptoms themselves could widen this variety. Recovery work is hard work, particularly in the presence of pernicious anxiety symptoms. As was learned during the first year of COVID-19, very few older people accessed digital resources [11] and others felt bewildered as to where to find support resources tailored to their needs [9]. A visual representation or blueprint showing how symptoms of anxiety can be connected and between strategies to manage or mitigate particularly pernicious ones could be helpful to practitioners, program developers, and older people, as they navigate through the complexities of post-COVID mentally healthy recovery work. Anxiety [34] and more isolated living [79] remain global health concerns, as is long-COVID [80].

Our speaking of a troubling trio shows the human side of anxiety management, with its trials and tribulations, and relatable imperfections. Everyday people need to be able to relate to research findings to talk about and learn from them [81]. We focus on symptoms and symptom-strategy relationships in our Network Analysis. This bird's eye view avoids a certain age or sex group from, for example, being singled out as anxiety laden. Our findings are meant to help older people have less self-conscious (not conspicuous) recovery work conversations with clinicians. Being a member of a population linked to COVID-19 [57] and mental health talk [82] is a social taboo. Ideally taking stock of peers' means of mitigating anxiety conjures taking stock of one's own. With responders as peer educators, shouldn't 'old as vulnerable' [83] be a taboo with similar social notoriety?

## Conclusions

Our study is the first study to offer evidence of what an optimal mentally healthy living structure might look like for older people, and as illustrated at a pivotal point in the COVID-19 pandemic. Mental health practitioners and program developers, and older people should find such discoveries intriguing. A combined blueprint of older Canadians' most pernicious anxiety symptoms and strategies to manage or mitigate them could be a helpful visual teaching tool that offers older people a better understanding of their own anxiety recovery work. Seeing what contemporaries' post-COVID-19 mentally healthy recovery work looks like might also offer a much-needed sense of hope for older people.

This study has limitations, however, including our reliance on data collected only in Canada, and at one point in time. We found associations between and among symptoms and coping behaviours. We cannot say with any degree of certainty that a troubling trio of anxiety symptoms evoke any one strategic action, nor a chain reaction in others.

While our observed network associations provide valuable insights, it would be important to investigate their directionality. For instance, does increasing resilience decrease feelings of tension or lack of control, or do these feelings prompt individuals to become more resilient? Thus, it is also crucial to consider external factors or other internal processes that might influence these associations. This includes life events, other coping strategies not in the network, or sociodemographic factors like personal income [23] and personality [27]. Research also tells us that older people have a capacity to better regulate negative emotions [27,84], even through the simplest domestic tasks [28] and by taking disrupted routines in stride [85]. Acknowledging and expressing negative emotions helped older Canadians persevere early in the pandemic [71].

How large-scale traumatic events unfold, and people's reactions and remedial resources at-hand can vary over time [86]. The COVID-19 pandemic has been described as sending unprecedented global socioeconomic and health-related shockwaves through everyday people's lives [1,2]. Its psychological repercussions are not fully realized [34,87]. In the first two years of the COVID-19 pandemic, older Canadians were living most at risk for falling gravely ill [8] and were living at arm's length from loved ones [88]. Contemporaries' anticipatory guidance for managing pernicious anxiety when transitioning into open social spaces during a future pandemic should resonate with other older Canadians and offer points of comparison for age-peers outside of Canada. Researchers stand to gain lived insights with which to interpret their own findings, both now and in future pandemic, and perhaps among other populations like older people acting to combat worrisome global climate change and to dampen their eco-anxieties [89].

## Acknowledgments

We thank our e-survey responders for their shared wisdom and resourcefulness.

## Author contributions

**Conceptualization:** Gail Low, Alex B. França, Gloria Gutman, Sofia von Humboldt.

**Data curation:** Gail Low, Zhiwei Gao, Hunaina Allana.

**Formal analysis:** Alex B. França.

**Funding acquisition:** Gail Low.

**Investigation:** Gail Low, Alex B. França.

**Methodology:** Gail Low, Alex B. França, Zhiwei Gao.

**Project administration:** Gail Low, Hunaina Allana, Donna M. Wilson, Anila Naz.

**Resources:** Gail Low, Gloria Gutman, Donna M. Wilson, Anila Naz.

**Software:** Alex B. França.

**Supervision:** Gail Low.

**Validation:** Gail Low, Gloria Gutman, Sofia von Humboldt.

**Visualization:** Gail Low, Alex B. França, Hunaina Allana.

**Writing – original draft:** Gail Low, Alex B. França.

**Writing – review & editing:** Zhiwei Gao, Gloria Gutman, Sofia von Humboldt, Hunaina Allana, Donna M. Wilson, Anila Naz.

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
