## [Decision Letter · Decision Letter 0]

18 Oct 2024

PMEN-D-24-00253

Anxiety symptoms and coping strategies used by older adults during COVID-19: A national e-study of linkages among and between them

PLOS Mental Health

Dear Dr. Low,

Thank you for submitting your manuscript to PLOS Mental Health. After careful consideration, we feel that it has merit but does not fully meet PLOS Mental Health’s publication criteria as it currently stands. Therefore, we invite you to submit a revised version of the manuscript that addresses the points raised during the review process.

Please note that we have only been able to secure a single reviewer to assess your manuscript. We are issuing a decision on your manuscript at this point to prevent further delays in the evaluation of your manuscript. Please be aware that the editor who handles your revised manuscript might find it necessary to invite additional reviewers to assess this work once the revised manuscript is submitted. However, we will aim to proceed on the basis of this single review if possible. 

In particular, the reviewer has requested further discussion about the findings and limitations of the study. Could you please carefully revise the manuscript to address all comments raised?

We look forward to receiving your revised manuscript.

Kind regards,

Helen Howard

Staff Editor

PLOS Mental Health

Additional Editor Comments (if provided):

Reviewers' comments:

Reviewer's Responses to Questions

**Comments to the Author**

1. Does this manuscript meet PLOS Mental Health’s publication criteria ? Is the manuscript technically sound, and do the data support the conclusions? The manuscript must describe methodologically and ethically rigorous research with conclusions that are appropriately drawn based on the data presented.

Reviewer #1: Yes

2. Has the statistical analysis been performed appropriately and rigorously?

Reviewer #1: Yes

3. Have the authors made all data underlying the findings in their manuscript fully available (please refer to the Data Availability Statement at the start of the manuscript PDF file)?

Reviewer #1: Yes

4. Is the manuscript presented in an intelligible fashion and written in standard English?

Reviewer #1: Yes

5. Review Comments to the Author

Reviewer #1: This study employed Network Analysis to create a visual representation of older Canadians’ symptoms of anxiety experiences and another of the linkages between their most pernicious symptoms and the strategies they used to cope with them. It provides a clear picture to show how older people managing or mitigating anxiety during the COVID-19 pandemic. Here are some suggestions for authors’ considerations: 1) Additional justification is required to substantiate the findings that depict the situation of older adults during the pandemic, given the absence of a 'control' group for comparison; 2) Given that COVID-19 is an unprecedented event, how the findings related to COVID-19 could inform scholars in the future and be applied to areas beyond the outbreak? 3) Please explain the distinctions between Network Analysis and traditional methods like logistic regression beyond just enhancing data visualization. 4) The conclusion section is too sketchy, limitations about the study should be added.

6. PLOS authors have the option to publish the peer review history of their article (what does this mean? ). If published, this will include your full peer review and any attached files.

**Do you want your identity to be public for this peer review?** For information about this choice, including consent withdrawal, please see our Privacy Policy .

Reviewer #1: No

---

## [Decision Letter · Decision Letter 1]

13 Jan 2025

PMEN-D-24-00253R1

Anxiety symptoms and coping strategies used by older adults during COVID-19: A national e-study of linkages among and between them

PLOS Mental Health

Dear Dr. Low,

Thank you for submitting your manuscript to PLOS Mental Health. After careful consideration, we feel that it has merit but does not fully meet PLOS Mental Health’s publication criteria as it currently stands. Therefore, we invite you to submit a revised version of the manuscript that addresses the points raised during the review process.

The manuscript has been evaluated by two reviewers, and their comments are available below.

The reviewers have raised a number of major concerns. In particular, they request improvements to the reporting of methodological aspects of the study, additional data, and further discussion.

Could you please carefully revise the manuscript to address all comments raised?

We look forward to receiving your revised manuscript.

Kind regards,

Helen Howard

Staff Editor

PLOS Mental Health

Additional Editor Comments (if provided):

Reviewers' comments:

Reviewer's Responses to Questions

**Comments to the Author**

1. If the authors have adequately addressed your comments raised in a previous round of review and you feel that this manuscript is now acceptable for publication, you may indicate that here to bypass the “Comments to the Author” section, enter your conflict of interest statement in the “Confidential to Editor” section, and submit your "Accept" recommendation.

Reviewer #1: All comments have been addressed

Reviewer #2: All comments have been addressed

2. Does this manuscript meet PLOS Mental Health’s publication criteria ? Is the manuscript technically sound, and do the data support the conclusions? The manuscript must describe methodologically and ethically rigorous research with conclusions that are appropriately drawn based on the data presented.

Reviewer #1: Yes

Reviewer #2: Partly

3. Has the statistical analysis been performed appropriately and rigorously?

Reviewer #1: Yes

Reviewer #2: N/A

4. Have the authors made all data underlying the findings in their manuscript fully available (please refer to the Data Availability Statement at the start of the manuscript PDF file)?

Reviewer #1: Yes

Reviewer #2: No

5. Is the manuscript presented in an intelligible fashion and written in standard English?

Reviewer #1: Yes

Reviewer #2: Yes

6. Review Comments to the Author

Reviewer #1: Thank you. I do not have further comments.

Reviewer #2: The conducted study is of great interest and, despite the time that has passed since the pandemic, remains relevant. Complex behavioral and emotional regular mechanisms involved in the population's reactions to the unprecedented conditions of the pandemic remain incompletely understood.

The chosen method of sample stratification and the authors' focus on the population of elderly respondents as a vulnerable group are of great value. Also valuable is the authors' practice-oriented approach to the problem under study and the desire to offer a visual tool for improving anti-stress mechanisms in the target audience.

The ethical aspect of the study, including personalized questionnaires with password access, as well as financial support for respondents, is commendable.

At the same time, a number of aspects of the work require clarification or supplementation.

1. The authors define the target audience for the application of the research results as elderly Canadians, but the article does not provide sociodemographic information about the sample for which the findings are applicable. What is the age dispersion, gender composition, economic level of the surveyed respondents, their employment and living conditions in the family? Without these data, the applicability of the study results is significantly limited.

2. In terms of the applicability of the study results, the fact that the authors do not provide data on anxiety symptoms and coping strategies, but on their interrelationships, also requires a more in-depth analysis in the article. The manuscript does not contain information on the prevalence or intensity of symptoms, but only on their interrelationships. There is also no data on the frequency or intensity of coping strategies. Thus, not only should the title of the article be reformulated with an emphasis on the "network model of anxiety symptoms and coping strategies", but the discussion can also be built on possible explanations for the associations obtained, rather than an analysis of the vulnerability of the target audience.

3. The explanations of the associations in the manuscript require more correct formulations. As the authors correctly clarify in the limitations section, causal relationships cannot be established in a cross-sectional study. In this case, the interpretation of the obtained data is performed by the authors with an implicit assumption that coping strategies determine the development of anxiety symptoms: “Our respondents’ remember resilience seemingly strongly shielded them from feeling restless, tense, and as if they had no control over their lives”; “older Canadians in this study were responding to a troubling trio of anxiety symptoms using multiple remedial strategies, seemingly with varying success”. Perhaps, it would be more effective to build a discussion through an analysis of effective and ineffective coping strategies, where ineffective strategies CAN provoke symptoms of (UNHEALTHY and HEALTHY) anxiety. Likewise, symptoms of (healthy) anxiety associated with the objectively existing threat of a pandemic CAN provoke the choice of ineffective coping strategies instead of effective ones, which reflects the maladaptive potential of the pandemic. 4. Smaller, but also important for improving the manuscript, comments can be taken into account by the authors:

A) the figures with the results of network analysis do not contain numerical values, which does not allow the reader to independently interpret the data. The strongest connections should be marked not only by the thickness of the lines, but also by numerical values. Otherwise, it is not obvious why the authors avoid discussing such anxiety symptoms as 5: Not control my worry and 10: Apprehensive. Unlike more traditional methods, such as factor and regression analysis, the proposed network analysis method does not have consensus "cutoff lines" and reliability levels, or they are not clearly enough presented in the article. Therefore, all connections (or at least half of the most significant of them) require explanation. Symptoms 5 and 10, according to the data in Figure 1, look more significant than symptom 9: No control over my life. According to Figure 3, their differences do not look significant enough. Similarly, according to Figures 3 and 4 for coping 4: Feeling anxiety is normal and coping 10: Challenging worries and anxious thoughts, it is necessary to provide an explanation why they are not included in the discussion (especially important in the context of comment #3 above).

B) The discrepancy in the number of respondents surveyed given in the abstract (1,327) and in the caption to Figure 1 (n = 1,224) requires an explanation. Perhaps there were socio-demographic characteristics of the respondents not included in the network analysis for constructing the figures?

C) An addition is required in the limitations section about the applicability of the results only to elderly residents of Canada, or the authors' arguments as to why the patterns found can be considered universal, and therefore in some part applicable (under what conditions) to wider target audiences

7. PLOS authors have the option to publish the peer review history of their article (what does this mean? ). If published, this will include your full peer review and any attached files.

**Do you want your identity to be public for this peer review?** For information about this choice, including consent withdrawal, please see our Privacy Policy .

Reviewer #1: No

Reviewer #2: No

---

## [Decision Letter · Decision Letter 2]

20 Mar 2025

Anxiety symptoms and coping strategies used by older adults during COVID-19: A national e-study of linkages among and between them

PMEN-D-24-00253R2

Dear Dr. Low,

We are pleased to inform you that your manuscript 'Anxiety symptoms and coping strategies used by older adults during COVID-19: A national e-study of linkages among and between them' has been provisionally accepted for publication in PLOS Mental Health.

Best regards,

Karli Montague-Cardoso

Executive Editor

PLOS Mental Health

Reviewer Comments (if any, and for reference):

Reviewer's Responses to Questions

**Comments to the Author**

1. If the authors have adequately addressed your comments raised in a previous round of review and you feel that this manuscript is now acceptable for publication, you may indicate that here to bypass the “Comments to the Author” section, enter your conflict of interest statement in the “Confidential to Editor” section, and submit your "Accept" recommendation.

Reviewer #2: All comments have been addressed

2. Does this manuscript meet PLOS Mental Health’s publication criteria ? Is the manuscript technically sound, and do the data support the conclusions? The manuscript must describe methodologically and ethically rigorous research with conclusions that are appropriately drawn based on the data presented.

Reviewer #2: Yes

3. Has the statistical analysis been performed appropriately and rigorously?

Reviewer #2: Yes

4. Have the authors made all data underlying the findings in their manuscript fully available (please refer to the Data Availability Statement at the start of the manuscript PDF file)?

Reviewer #2: Yes

5. Is the manuscript presented in an intelligible fashion and written in standard English?

Reviewer #2: Yes

6. Review Comments to the Author

Reviewer #2: The authors substantially revised the manuscript and took into account all the recommendations of the reviewer.

7. PLOS authors have the option to publish the peer review history of their article (what does this mean? ). If published, this will include your full peer review and any attached files.

**Do you want your identity to be public for this peer review?** For information about this choice, including consent withdrawal, please see our Privacy Policy .

Reviewer #2: No
